# Modeling the Impact of COVID-19 on Air Quality in Southern California: Implications for Future Control Policies

Zhe Jiang[1, 2†], Hongrong Shi[1*†], Bin Zhao[3*], Yu Gu[4], Yifang Zhu[5], Kazuyuki Miyazaki[6], Xin Lu[7], Yuqiang Zhang[8], Kevin W. Bowman[4,6], Takashi Sekiya[9], Kuo-Nan Liou[4]

[1] Key Laboratory of Middle Atmosphere and Global Environment Observation, Institute of Atmospheric Physics, Chinese Academy of Sciences, Beijing, China
  [2] Carbon Neutrality Research Center, Institute of Atmospheric Physics, Chinese Academy of Sciences, Beijing, China
  [3] Pacific Northwest National Laboratory, Richland, WA, USA
  [4] Joint Institute for Regional Earth System Science and Engineering and Department of Atmospheric and Oceanic Sciences,
University of California, Los Angeles, CA, USA
  [5] Institute of Environment and Sustainability, University of California, Los Angeles, CA, USA
  [6] Jet Propulsion Laboratory, California Institute of Technology, Pasadena, CA, USA
  [7] Petrochina North China Gas Marketing Company, Beijing, China
  [8] Nicholas School of the Environment, Duke University, Durham, NC, USA
[9] Japan Agency for Marine-Earth Science and Technology, Yokohama, Japan

Correspon*dence to*: Hongrong Shi (shihrong@mail.iap.ac.cn) and Bin Zhao (bin.zhao@pnnl.gov)

† Zhe Jiang and Hongrong Shi contributed equally to this paper.

**Abstract.** In response to the Coronavirus Disease 2019 (COVID-19), California issued statewide stay-at-home orders,
bringing about abrupt and dramatic reductions in air pollutant emissions. This crisis offers us an unprecedented opportunity to evaluate the effectiveness of emission reductions on air quality. Here we use the Weather Research and Forecasting model with Chemistry (WRF-Chem) in combination with surface observations to study the impact of the COVID-19 lockdown measures on air quality in southern California. Based on activity level statistics and satellite observations, we estimate the sectoral emission changes during the lockdown. Due to the reduced emissions, the population-weighted concentrations of
fine particulate matter ($PM_{2.5}$) decrease by 15% in southern California. The emission reductions contribute 68% of the $PM_{2.5}$ concentration decrease before and after the lockdown, while meteorology variations contribute the remaining 32%. Among all chemical compositions, the $PM_{2.5}$ concentration decreases due to emission reductions is dominated by nitrate and primary components. For $O_3$ concentrations, the emission reductions cause a decrease in rural areas but an increase in urban areas; the increase can be offset by a 70% emission reduction in anthropogenic volatile organic compounds (VOC). These findings
suggest that a strengthened control on primary $PM_{2.5}$ emissions and a well-balanced control on nitrogen oxides and VOC emissions are needed to effectively and sustainably alleviate $PM_{2.5}$ and $O_3$ pollution in southern California.

# 1 Introduction

Anthropogenic emissions from various emission sources, including transportation, industrial, agricultural, residential, and commercial sectors, contribute to California's long-existing air pollution problems (e.g., Shirmohammadi et al., 2016; Hong et al., 2015; Warneke et al., 2013). The major pollutants include, but are not limited to, fine particulate matter ($PM_{2.5}$), nitrogen dioxide ($NO_2$), sulfur dioxide ($SO_2$) and ozone ($O_3$). Exposure to these pollutants has been correlated with an increased rate of morbidity and mortality (Wang et al., 2019). Mitigating the adverse effects of air pollution by reducing air pollutant emissions from major sectors has been and will continue to be a major public policy challenge. However, the effect of emission reductions from various sources on air quality improvement is subject to substantial uncertainties, because such effect cannot be directly measured and because the atmospheric chemistry processes are highly nonlinear and complicated (Zhao et al., 2019b; Zhao et al., 2015; Chen et al., 2013). The recent Coronavirus Disease 2019 (COVID-19) pandemic provides an unprecedented opportunity for a more robust understanding of the environmental impacts brought by the emission reductions.

More than 200 countries and territories around the world have reported a total of about 53 million confirmed cases of the coronavirus COVID-19 that originated from Wuhan, China, and a death toll of more than 1300K (World Health Organization, 2020). California is one of the most affected states in the United States (U.S.) partly because its poor air quality makes Californians more susceptible to infectious diseases such as COVID-19 (Bashir et al., 2020; Chiara Copat, 2020). In response to the emergence of COVID-19, statewide stay-at-home orders and related actions (e.g., closure of nonessential businesses) took effect on March 19, 2020 in California. These orders are expected to reduce vehicle traffic and industrial activities, thereby changing the air pollutant emissions and air quality in the state. It is essential to obtain a high-spatiotemporal-resolution estimation of air pollution for better understanding of the atmospheric impacts caused by changes in anthropogenic activity associated with the COVID-19 pandemic.

A number of studies emerged soon after the start of the COVID-19 pandemic and the subsequent lockdown to assess the impact of the pandemic on air quality over various regions around the world. For example, Archer et al. (2020) compared the observed concentrations at all available ground monitoring sites in U.S. between April of 2020 and the prior five years (2015–2019) and found statistically significant decreases in $NO_2$ concentrations at more than 65% of the monitoring sites, with an average drop of 2 ppb. Pan et al. (2020) compared the surface air quality monitoring data in California during the period 20 March–5 May in 2020 with those in 2015–2019 and found that the $PM_{2.5}$ in 2020 exhibited a notable decrease which could result from emission reductions associated with the COVID-19 lockdown. Similar findings, i.e., reduced $PM_{2.5}$ and $NO_2$ concentrations are also reported for China (e.g., Chu et al., 2020; Le et al., 2020; Liu et al., 2020; Marlier et al., 2020; Shi and Brasseur, 2020; Miyazaki et al., 2020b), India (e.g., Pathakoti et al., 2020; Sharma et al., 2020), and Europe (e.g., Chen et al., 2020; Menut et al., 2020; Sicard et al., 2020; Ordóñez et al., 2020) based on surface and/or satellite observations. For $O_3$, the concentrations either increased or slightly decreased during the pandemic, depending on regions (Bekbulat et al., 2020; Huang et al., 2020; Pan et al., 2020; Zhao et al., 2020). Most of the above studies, however, are

limited to comparing observations with and without lockdown measures, which correspond to different time periods under different meteorological conditions.

Meteorology plays significant roles in air pollution formation, transport, deposition and transformation (Wang et al., 2020a), which is a very important factor that affects concentrations of $O_3$ and $PM_{2.5}$ (Stewart et al., 2017). The changes in air quality due to meteorological variations may obscure the effects of emission changes during the COVID-19 lockdown. Using the

Community Multi-scale Air Quality model, Wang et al. (2020a) showed that the benefits of emission reductions were overwhelmed by adverse meteorology over the North China Plain and severe air pollution events were thus not avoided. Goldberg et al. (2020) reported that meteorological patterns were especially favorable for low $NO_2$ in much of the United States in spring 2020, complicating comparisons with spring 2019; the meteorological variations between years can cause ~15% difference in monthly mean column $NO_2$. In view of this, modelling approach is necessary to accurately assess the

impact of lockdown measures by excluding the possible effects of meteorological conditions and to examine the possible mechanisms responsible for the changes in the air pollutant concentrations. In addition, while previous studies have evaluated the air quality changes in different regions due to the emission reductions associated with the COVID-19 lockdown, it remains unclear how the COVID-19 induced emission reductions and the concurrent meteorological variations influence air quality in California.

The objective of this study is to investigate the air quality impact of the emission reductions in southern California in association with COVID-19 by employing high-resolution atmospheric modelling in combination with surface observations. Based on the statistics of activity levels together with constraints from satellite observations, we estimate the sectoral emission changes during the COVID-19 lockdown. We then conduct model simulations using the Weather Research and Forecasting model with Chemistry (WRF-Chem) for the periods before and during the COVID-19 lockdown to investigate

the effects of reduced emissions and meteorology on air pollution, respectively. Understanding how air quality responds to the emission reductions during COVID-19 pandemic will provide important insight into the future development and optimization of effective air pollution control strategies in southern California.

## 2 Method and Data

### 2.1 Model configuration and emission estimation

We simulate the impact of COVID-19 lockdown measures on air quality using the WRF-Chem version 3.9.1, which considers highly nonlinear and complex meteorological and atmospheric chemistry processes. The simulation period is February 18 to April 23, 2020, which includes about one month before and after the California governor issues the stay-at-home (lockdown) order on March 19 (Pan et al., 2020). We apply the model to two nested domains: Domain 1 covers the western United States and its surrounding areas at a 12 km×12 km horizontal resolution; Domain 2 covers California with a

4 km×4 km resolution (Fig. 1). We focus our analysis on southern California (the red rectangle in Fig. 1), the largest metropolitan area in California which is significantly affected by the lockdown measures. We classify model grids in

southern California into "urban" and "rural" areas to facilitate the analysis of $O_3$ simulation results. To be classified as "urban", an area in the U.S. needs to have a population density of 1,000 people per square mile (Ratcliffe et al., 2016), i.e., about 6000 people per 4 km×4 km model grid. As we focus our analysis on southern California, one of the most densely populated areas in the U.S., we use a higher population density threshold of 30,000 people per model grid to better distinguish areas with different photochemistry regimes (Fig. S1). We employ an extended Carbon Bond 2005 (CB05) (Yarwood et al., 2005) with chlorine chemistry (Sarwar et al., 2008) coupled with the Modal for Aerosol Dynamics in Europe/Volatility Basis Set (MADE/VBS) (Wang et al., 2015a; Ahmadov et al., 2012). MADE/VBS uses a modal aerosol size representation and an advanced secondary organic aerosol (SOA) module based on the VBS approach. The aqueous-phase chemistry is based on the AQChem module used in the Community Multiscale Air Quality (CMAQ) model (Wang et al., 2015a). The chemical initial and boundary conditions were extracted from the output of the Whole Atmosphere Community Climate Model (WACCM) (Marsh et al., 2013). A 6-day spin-up period is used to minimize the influence of initial conditions on simulation results. The vertical resolution, meteorological initial and boundary conditions, and physical options are the same as our previous modeling studies based on WRF-Chem for California (Zhao et al., 2019a; Wang et al., 2020b; Shi et al., 2019).

We obtain anthropogenic emissions in California without the influence of COVID-19 lockdown measures from the California Air Resources Board (CARB) for 2012 that is the latest year in which the data are available (California Air Resources Board, 2018). We scale the 2012 emissions to the 2020 levels by employing the relative changes for 2012–2018 in California from the "NEI trend report" (US Environmental Protection Agency, 2018a) and assuming that the trends continued during 2018–2020. The anthropogenic emissions outside California are derived from the National Emission Inventory (US Environmental Protection Agency, 2018b) in 2011 and are scaled to 2020 following the same method. The biogenic, wind-blown dust, sea-salt, and wildfire emissions are calculated online in WRF-Chem, as detailed in our previous studies (Zhao et al., 2019a; Wang et al., 2020b; Shi et al., 2019).

In our baseline simulation ("Base" scenario in Table 1), we use the above emission inventories. To evaluate the effect of the COVID-19 response actions, we conduct another simulation ("Lockdown" scenario in Table 1) in which the CARB anthropogenic emission inventory after March 19 is adjusted to account for the emission changes due to the COVID-19 lockdown. Because of the lack of detailed emission data which often take years to update, we rely on a number of key activity indicators to estimate the sector-specific relative changes in anthropogenic emissions (as summarized in Table 2), which are subsequently evaluated against satellite-derived emission estimate. For the transportation sector, we separately estimate the reduction rates for onroad, off-road, and aircraft emissions due to the COVID-19 lockdown. Specifically, we assume the reduction rates in gasoline and diesel vehicle emissions in the onroad sector to be the same as the reduction rates in gasoline and diesel production from the pre-lockdown period to the lockdown period, as documented by California Energy Commission's "Weekly Fuels Watch Reports" (California Energy Commission, 2020b). We then estimate the reduction rates in total emissions from the onroad sector based on the relative fractions of gasoline and diesel vehicle emissions reported by the CARB emission inventory. Since the off-road sector involves few gasoline vehicles, we assume the reduction rates in

off-road emissions to be the same as the reduction rate in diesel production. For the aircraft sector, we assume the reduction rates in aircraft emissions to be the same as the reduction rate in jet fuel production from the "Weekly Fuels Watch Reports" (California Energy Commission, 2020b). The changes in emissions from the industrial, residential, and commercial sectors are assumed to be proportional to the changes in electricity consumption by the corresponding sector, as summarized in the "Energy Insights Reports" of the California Energy Commission (California Energy Comission, 2020a). The changes in emissions from power plants are estimated as a function of the total electricity demand in California (California Energy Comission, 2020a). We also checked the emission change of power plants measured by the Continuous Emission Monitoring System (CEMS). There are certain differences between the emission reduction rates estimated based on the CESM and electricity demand, but the difference only results in a less than 1% difference in the total emissions of any pollutant (from 0.05% to 1%), which is expected to have a limited effect on the simulation results of mean air pollutant concentrations in southern California (see details in the Supplementary text and Fig. S3). Having estimated the emission changes using the preceding bottom-up method, in order to prove the reliability of our bottom-up emissions, we compare the changes in nitrogen oxides ($NO_x$) emissions with a top-down satellite-based emission inventory—an extended calculation of the Tropospheric Chemistry Reanalysis version 2 (TCR-2) (Miyazaki et al., 2020a). This data product has been obtained from the assimilation of multiple satellite measurements of ozone, CO, $NO_2$, $HNO_3$, and $SO_2$ from the OMI (Ozone Monitoring Instrument), TROPOMI (TROPOspheric Monitoring Instrument), MLS (Microwave Limb Sounder), and MOPITT (Measurement Of Pollution In The Troposphere) satellite instruments. The reanalysis calculation for the COVID-19 time period was conducted at $0.56°$ horizontal resolution using a global chemical transport model MIROC-CHASER (Watanabe et al., 2011) and an ensemble Kalman filter technique that optimizes chemical concentrations of various species and emissions of $NO_x$, $SO_2$, and CO. The extended reanalysis data for 2020 have already been used by Miyazaki et al. (2020b) to study air quality response to the Chinese COVID-19 lockdown measures. Here we use the $NO_x$ emission product which has a sufficiently high quality on the spatiotemporal scales of interest for this study. Using this product, we first calculate $NO_x$ emissions in a hypothetical scenario without considering the COVID-19 effect. Here the hypothetical emission trend in 2020 is averaged from those trends from the top-down NOx emission inventory in the prior years (2017-2019). We subsequently quantify the emission changes due to the COVID-19 using the difference between the hypothetical and real-world emissions (see details in Fig. 2). The estimated $NO_x$ reduction ratio induced by the COVID-19 lockdown measures averaged during March 19 to April 23 in southern California is 27.2% based on the top-down method, which is in good agreement with the 28.3% (see Table 2) reduction estimated based on the bottom-up method. That said, we acknowledge that, since more detailed data to support a more accurate estimation are not yet available, the estimates of the sector-specific relative changes in emissions inevitably involve some degree of uncertainty, which can be improved in the future work.

**2.2 Observational data and model evaluation**

We use a series of meteorology and air quality observations to evaluate the model performance and help analyze the influence of the COVID-19 lockdown. For meteorology, we use observational data obtained from the National Climatic Data

Center (NCDC), where hourly or 3-hour observations of wind speed at 10 m (WS10), temperature at 2 m (T2), and water vapor mixing ratio at 2 m (Q2) are available for 82 sites distributed southern California (the red rectangle in Fig. 1). We compare the WRF-Chem meteorological simulations with these measurements and apply a number of statistical indices defined in Emery et al. (2001) to quantitatively evaluate the model performance, as summarized in Table 3. In general, the model simulations agree fairly well with surface meteorological observations. The performance statistics for WS10, T2 and Q2 are all within the benchmark ranges proposed by Emery et al. (2001).

For air quality, we achieve hourly observations of $PM_{2.5}$, $O_3$, $NO_2$ and $SO_2$ from CARB (California Air Resources Board, 2020) and use them to evaluate the air quality simulations of WRF-Chem (see the Results and Discussion section). The observational data are available at 42 sites for $PM_{2.5}$, 63 sites for $O_3$, 48 sites for $NO_2$, and 12 sites for $SO_2$, in southern California (the red rectangle in Fig. 1) during the simulation period. We do not evaluate the model performance in simulating the chemical composition of $PM_{2.5}$ because the composition data from major observational networks had not been available by the time we completed the present study. Nevertheless, our previous studies using almost the same model configurations showed a fairly good agreement with $PM_{2.5}$ composition observations during January, April, July, and October, 2012 (Zhao et al., 2019a; Wang et al., 2020b).

## 3 Results and Discussion

### 3.1 Evaluation of the simulated results with surface observations

In this study, we simulated the major air pollutants using WRF-Chem under two scenarios, Base and Lockdown (Table 1). To evaluate the model performance with regard to the temporal variations in air pollutants, we compared the simulated concentrations of $PM_{2.5}$, maximum daily 8-h average (MDA8) $O_3$, $NO_2$ and $SO_2$ with observational data from CARB in southern California.

Before the COVID-19 lockdown (February 18 to March 18), results from model simulation under the Base scenario ($Pre_{Base}$) capture the magnitude and temporal evolution of the four key air pollutants reasonably well, with normalize mean biases (NMBs) of 11.7%, 4.5%, -14.4% and 7.8% for $PM_{2.5}$, MDA8 $O_3$, $NO_2$, $SO_2$, respectively (Fig. 3). During the COVID-19 lockdown period (March 19 to April 23), compared to the simulations for the Base scenario ($Post_{Base}$) which overestimates the surface concentrations with NMBs of 28.1%, 1.6%, 21.4% and 39.2% for $PM_{2.5}$, MDA8 $O_3$, $NO_2$, $SO_2$, respectively, the simulated results using the adjusted emission inventory ($Post_{Lockdown}$) not only agree better with surface observations for all the four air pollutants (with NMBs of 10.6%, 1.0%, -12.6% and -13.1% for $PM_{2.5}$, MDA8 $O_3$, $NO_2$, $SO_2$, respectively), but also show generally closer NMBs to those during the pre-lockdown period (Fig. 3). The improvement in model performance is observed for both urban and rural areas. In the urban areas, the NMB for $PM_{2.5}$ drops from 25.8% under the Base scenario to 3.9% under the Lockdown scenario, getting closer to the NMB of 4.0% during the pre-lockdown period. The corresponding NMB in rural areas drops from 29.7% to 15.1%, also getting closer to 17.8% during the pre-lockdown period (Figs. 3e,g). Regarding MDA8 $O_3$, although the differences between the Base and Lockdown scenarios are quite small (Fig.

3b), the NMB is slightly improved from -1.5% ($Post_{Base}$) to -0.2% ($Post_{Lockdown}$) in urban areas and from 3.2% to 1.5% in rural areas (Figs. 3f,h).

Subsequently, we evaluated the spatial distributions of simulated $PM_{2.5}$ and MDA8 $O_3$ concentrations using observational data averaged during the pre-lockdown and lockdown periods in southern California (Fig. 4). Figure S4 shows the scattergrams of the simulated and observed monthly average $PM_{2.5}$ and MDA8 $O_3$ concentrations in southern California. The Base scenario can simulate the spatial patterns of $PM_{2.5}$ and MDA8 $O_3$ reasonably well (Figs. 4a-b and d-e), but it overestimates the observations of $PM_{2.5}$ concentrations during the lockdown period ($Post_{Base}$, Figs. 4b and S4b). The simulated distributions of $PM_{2.5}$ concentrations under the Lockdown scenario ($Post_{Lockdown}$) match the observations better than those for the Base scenario ($Post_{Base}$) (Figs. 4b-c and S4b-c); the hot spots occurring over the Los Angeles County become less polluted and more consistent with the surface observations after considering the emission reductions associated with the COVID-19 lockdown (Figs. 4b-c).

## 3.2 Effects of anthropogenic emission reductions and meteorology conditions on air pollutants

Both observations and simulations in Fig. 3 show significant changes in air pollutant concentrations during the COVID-19 lockdown relative to the pre-lockdown period, resulting from a combination of emission reductions and meteorology variations. Our model simulations allow us to quantify the relative contributions of these two factors. Figures 5a-f illustrate population-weighted concentrations of simulated $PM_{2.5}$ components, $NO_2$, $SO_2$, MDA8 $O_3$ in southern California, and MDA8 $O_3$ over the urban and rural areas of southern California under the Base and Lockdown scenarios. We use population-weighted concentrations because they are more relevant to the health impacts of air pollutants ($PM_{2.5}$ and $O_3$), the mitigation of which is an ultimate goal of controlling air pollution. Figure S5 shows the mean concentrations of simulated $PM_{2.5}$ components, MDA8 $O_3$, $NO_2$, and $SO_2$ in southern California.

The simulations of the Base and Lockdown scenarios during the lockdown period ($Post_{Base}$ and $Post_{Lockdown}$) have the same model configurations and inputs (same large-scale meteorological conditions) except for different emission inventories. The concentration differences between the two scenarios during the lockdown period ($Post_{Lockdown}-Post_{Base}$) represent the effect of anthropogenic emission reductions. Strictly speaking, while the large-scale meteorological fields are the same in $Post_{Base}$ and $Post_{Lockdown}$, the different emission inputs could cause small differences in regional meteorology fields through the interactions between air pollutants and meteorology. Such a meteorology perturbation is considered to be part of the emission reduction effect because it is fundamentally caused by emission reductions. The simulations of the Base scenario during the lockdown and pre-lockdown periods ($Post_{Base}$ and $Pre_{Base}$) both use the emission inventories without considering the COVID-19 induced emission reductions. The differences between $Post_{Base}$ and $Pre_{Base}$ can be regarded as the impact of meteorology variations. Here our intention is to examine the relative contribution of meteorological variations to the population-weighted air pollutant concentrations before and after the lockdown, instead of the changes relative to the climatological conditions. However, we acknowledge that it is more meaningful and informative to assess the meteorological effect by conducting ensemble simulations over multiple years or use multi-year averaged meteorological conditions to serve

as a reference state (Le et al., 2020), which warrants further studies in the future. Figures 6 and 7 further show the spatial
distribution of the concentration changes caused by anthropogenic emission reductions and meteorology variations.

The simulated population-weighted $NO_2$ concentrations during the lockdown decrease by 4.3 ppb (from 10.7 to 6.4 ppb)
relative to the pre-lockdown period, of which the anthropogenic emission reductions and meteorology conditions contribute
2.4 ppb (56%) and 1.9 ppb (44%), respectively (Fig. 5b). The decrease in $NO_2$ concentrations as a result of the anthropogenic
emission reductions (27%) is similar to the reduction ratio in $NO_x$ emission (28%), indicating that the $NO_x$ emission
reductions can be almost fully transferred to ambient concentrations. According to our emission estimation, over 80% of the
$NO_x$ reductions is attributed to the substantially lowered traffic intensity due to the stay-at-home order. Note that the soil $NO_x$
emissions are not taken into account in our WRF-Chem simulation. According to Guo et al. (2020), the total soil $NO_x$
emissions in California account for only about 1.1% of the state's total anthropogenic $NO_x$ emissions (California Air
Resources Board, 2017). The soil $NO_x$ emissions in southern California are even generally lower compared with other parts
of the state (Guo et al., 2020). Since our study focuses on the impact of anthropogenic emission reductions on air quality
during the COVID-19 lockdown period, the absence of soil $NO_x$ emissions has little impact on our main results and will not
change the main findings of this study. The population-weighted concentrations of $SO_2$ also show a decreasing trend (Fig.
5c). Compared with $NO_2$, the decrease in $SO_2$ concentrations due to emission reductions is smaller (17%), partly because
power generators and heavy industry (the main sources of $SO_2$) are less affected by the COVID-19 lockdown (see Table 2).

Coinciding with the decrease in $NO_2$ and $SO_2$, the simulated population-weighted $PM_{2.5}$ concentrations decrease by 1.8
μg/m$^3$ from 8.7 μg/m$^3$ during the pre-lockdown period (Pre$_{Base}$) to 6.9 μg/m$^3$ during the lockdown period (Post$_{Lockdown}$). The
emission reductions contribute 1.2 μg/m$^3$ (67%) of the above decrease, which translates into a 15% reduction in population-
weighted $PM_{2.5}$ concentrations from the levels without the lockdown (i.e., Post$_{Base}$) (Fig. 5a). The decrease occurs almost
everywhere across the domain (Fig. 6a), consistent with the results in the last section that $PM_{2.5}$ concentrations are lowered
in both urban and rural areas as a result of the emission reductions (Figs. 3e,g). The concentration decrease is higher in urban
areas than in rural areas (Figs. 6a and 3e,g), with the most significant decline occurring in urban areas of the Los Angeles
County (Fig. 6a). In contrast, the meteorology variations can increase the $PM_{2.5}$ concentrations in some regions (mainly the
inland regions) and decrease them in others (mainly the coastal regions) (Fig. 6b). The net effect is to reduce the population-
weighted concentration by 0.6 μg/m$^3$ since the concentration decrease happens to occur in more densely populated regions
(Fig. 5a).

The concentrations of $PM_{2.5}$ are affected by emissions of multiple pollutants through both primary emissions and chemical
reactions. To further explore the reasons behind the $PM_{2.5}$ concentration changes, we examine the changes in individual
chemical components, as shown in Fig. 5a and Fig. 7. Following the emission changes (from Post$_{Base}$ to Post$_{Lockdown}$), all
major $PM_{2.5}$ components experience a concentration decrease almost throughout the domain (Fig. 7), since the emissions of
essentially all pollutants are reduced to some extent due to the lockdown measures (Table 2). The population-weighted
concentrations of nitrate decrease the most (0.42 μg/m$^3$), followed by "Others" (0.32 μg/m$^3$, including all other components
besides the key components listed here), organic matter (OM, 0.16 μg/m$^3$), ammonium (0.15 μg/m$^3$), black carbon (BC, 0.10

μg/m$^3$), and sulfate (from 0.07 μg/m$^3$) (Fig. 5a). The largest decrease in nitrate is tied to the substantial reduction in NO$_x$ emissions, which is further explained by a larger reduction ratio in transportation emissions (by 30–70%) compared with other emission sources (Table 2). In addition, the decreases in "Others", EC, and primary OM (a fraction of the total OM) are attributable to the reductions in primary PM$_{2.5}$ emissions. In our emission estimates, the sector-specific relative emission changes of EC, primary OM, and "Others" are assumed to be the same as the total primary PM$_{2.5}$, as summarized in Table 2. For the total emissions of all sectors, the reduction in EC, primary OM, and "Others" are 22.7%, 15.8%, and 13.5%, respectively, slightly different from the reduction in total primary PM$_{2.5}$ since different chemical components have different sectoral distributions. The overall concentration decrease in these primary chemical components even exceeds that of nitrate; this clearly indicates an important role of primary PM$_{2.5}$ components in improving PM$_{2.5}$ air quality during the lockdown period, although the primary PM$_{2.5}$ emissions have only been reduced by 15%.

The simulated population-weighted O$_3$ concentrations increase noticeably from 38 ppb in the pre-lockdown period (Pre$_{Base}$) to 42 ppb (Post$_{Lockdown}$) during the lockdown, and the effects of meteorological changes (i.e. Post$_{Base}$−Pre$_{Base}$) play a dominant role in the variation of O$_3$. The O$_3$ level is strongly affected by ambient conditions like temperature and solar radiation (Wang et al., 2015b). As the temperature gets warmer and the radiation gets stronger over time, the O$_3$ concentrations are elevated in most areas during the COVID-19 lockdown, compared to the pre-lockdown period (Fig. 6d). The emission reductions cause an O$_3$ decrease in rural areas but a slight increase in the urban areas (Fig. 6c and Figs. 3f,h), which is consistent with previous findings (Zhao et al., 2019a; Wang et al., 2020b; Martien et al., 2003; Qin et al., 2004). In urban areas where NO$_x$ emissions are high, a volatile organic compounds (VOC)-limited regime is seen, while in rural areas, a NO$_x$-limited regime is observed (Martien et al., 2003; Qin et al., 2004). It follows that the decrease in NO$_x$ emissions leads to opposite changes in O$_3$ concentrations in urban and rural areas. The increase and decrease in different areas largely offset each other, resulting in a negligible change in population-weighted O$_3$ concentrations (0.07 ppb) (Fig. 5d) and a slight decrease in area-averaged O$_3$ concentrations over the modelling domain (0.77 ppb) (Figs. 6c and S5b). Last but not least, the small sensitivity of O$_3$ to emission reductions is also partly explained by the fact that 75% of the ambient O$_3$ concentration is background O$_3$ (Zhao et al., 2019a; Wang et al., 2020b).

**3.3 Effects of anthropogenic NO$_x$ and VOC emission reductions on ozone concentration**

Our modelling results showed an increase in O$_3$ in urban areas due to the emission reductions in association with the lockdown during the COVID-19 pandemic. The O$_3$ concentrations are most significantly affected by emissions of NO$_x$ and VOC (Stewart et al., 2017). To further explore the drivers of the O$_3$ changes and potential approaches to effectively reduce O$_3$ concentrations, we conduct three sensitivity experiments involving NOx and VOC emission perturbations, as summarized in Table 1. Figure 8 illustrates population-weighted concentrations of simulated PM$_{2.5}$ components and MDA8 O$_3$ after the COVID-19 lockdown under these sensitivity scenarios. Figure 9 shows the spatial distribution of the differences in MDA8 O$_3$ between the sensitivity scenarios and the Base scenario. The first sensitivity experiment is the VOC1.0 scenario which is

the same as "Lockdown" except that the VOC emissions are kept at the level of the "Base" scenario (Table 1). This scenario, in combination with the Base and Lockdown scenarios, can be used to evaluate the response of $O_3$ concentrations if the COVID-19 induced emission reductions of $NO_x$ and VOC were implemented in sequence. Without the control of VOC emissions in VOC1.0 (Fig. 9a), the increase in urban $O_3$ concentration relative to the Base scenario becomes larger than the Lockdown scenario (Fig. 6c). This confirms our analysis in the last section that the $NO_x$ emission control elevates urban $O_3$ concentrations under the VOC-limited regime and meanwhile indicates that the VOC control is conducive to $O_3$ decrease. To assess the potential effects of strengthened $NO_x$ and VOC control measures, we conduct two other sensitivity experiments named $NO_x$0.3 and VOC0.3, which are the same as "Lockdown" except that the anthropogenic $NO_x$ (for the $NO_x$0.3 scenario) and VOC (for the VOC0.3 scenario) emissions are further reduced to 30% of those in the "Base" scenario. As a 70% reduction is close to the maximum reductions in $NO_x$ and VOC emissions that could be achieved through the full implementation of technologically and economically feasible control measures (Amann et al., 2020), we select an emission ratio of 0.3 (70% reduction) to represent the potential impact of highly stringent control policies in the future. Figs. 8a,b show that strengthened $NO_x$ control further reduces the population-weighted concentrations of $PM_{2.5}$, while further reduction of anthropogenic VOC helps to decrease the concentration of MDA8 $O_3$. Differences in $O_3$ concentration clearly illustrate different spatial distribution patterns for urban and suburb areas (Figs. 9b, c). For the suburbs with high $O_3$ values, reducing anthropogenic $NO_x$ and VOC is conducive to the decline of $O_3$ (Fig. 8d). For urban areas, however, strengthened control with anthropogenic $NO_x$ reduced by 70% ($NO_x$0.3) results in even more $O_3$ increase in the central urban area (Figs. 9b and 8c). Amplified ozone pollution has also been reported by Sicard et al. (2020) based on their observational studies in four Southern European cities and Wuhan, China associated with $NO_x$ reductions in response to COVID-19. To control $O_3$ concentrations in urban areas, VOC control may be an effective method. While a $NO_x$ emission reduction might cause an increase in $O_3$ concentration, a VOC reduction generally leads to a monotonous reduction of $O_3$ concentrations regardless of the $O_3$ formation regime, as indicated by the classical $O_3$ EKMA isopleth (Figure 6-1 of National Research Council (1991) or Figure 3.2.1 of Donahue (2018)) as well as some recent studies in southern California (Fujita et al., 2013; Collet et al., 2018; Qian et al., 2019). We find that in the VOC0.3 scenario, there is almost no $O_3$ concentration increase relative to the Base scenario, in contrast to a significant urban $O_3$ increase in the Lockdown scenario (Fig. 9c). This means that a 70% reduction in anthropogenic VOC can offset the increases in $O_3$ caused by the 28.3% $NO_x$ reduction during the lockdown. Note that we are specifically looking at the extent of VOC emission reductions that are needed to offset the 28.3% $NO_x$ reduction caused by the lockdown, which minimizes the complexity due to the nonlinear $O_3$ responses when the $NO_x$ emissions are changing simultaneously. Furthermore, Wang et al. (2019) found that 75% of the average $O_3$ concentration in California was due to distant emissions outside the western United States. Many other studies also revealed that the background $O_3$ dominates over the contribution from local emissions in the western U.S. (Huang et al., 2015; Oltmans et al., 2008; Fiore et al., 2014; Emery et al., 2012; Zhang et al., 2011). Therefore, cooperating with other regions and countries in emission reductions may be another way to control $O_3$ in urban areas of the southern California.

## 4 Conclusion and policy implications

In this study, we investigated the air quality impact of the emission reductions in southern California in association with COVID-19 by employing WRF-Chem to conduct high-resolution atmospheric modeling during February 18 to April 23, 2020.

Based on the statistics of activity levels, we first adjusted the emission inventory considering the emission reductions during the COVID-19 lockdown. The adjusted emission inventory is shown to be consistent with the emission inventory based on satellite observations. The simulated magnitude and temporal evolution of the concentrations of the key air pollutants, including $PM_{2.5}$, $NO_2$, $SO_2$, and MDA8 $O_3$ using the adjusted emission inventory agree better with surface observations than simulation results without considering the COVID-19 induced emission reductions. Due to the reduced emissions, the population-weighted concentrations of $NO_2$ and $PM_{2.5}$ decreased by 27% and 15%, respectively, in southern California in the five weeks after the stay-at-home orders. Emission reductions and meteorological variations contributed about two-thirds and one-third, respectively, to the total decrease in population-weighted $PM_{2.5}$ concentrations before and after the lockdown. For $O_3$ concentration, however, the COVID-19 related emission reductions caused a decrease in suburb areas but a slight increase in the urban areas. In order to further explore the effects of anthropogenic $NO_x$ and VOC emission reductions on $O_3$ concentration, we conducted sensitivity experiments by strengthening VOC and $NO_x$ controls. Our results showed that strengthened control with $NO_x$ reduced by 70% ($NO_x$0.3) results in even more $O_3$ increase in the central urban area and anthropogenic VOC control may be an effective method to reduce $O_3$ concentrations in urban areas. A 70% reduction in anthropogenic VOC can effectively offset all the increases in $O_3$ caused by $NO_x$ reduction during the lockdown.

Using the COVID-19 as an unprecedented experiment with substantial emission reductions from multiple sectors, especially transportation, this study helps to elucidate the complex and nonlinear response of chemical compositions to air pollution control measures and thus provides important insight into the development and optimization of effective air pollution control strategies in southern California. We find that the reduced $NO_x$ emission (~28%) has been almost fully transferred to the reduction in ambient concentration of $NO_2$ (~27%). This further translates into a remarkable reduction in nitrate, which makes the largest contribution to $PM_{2.5}$ concentration decrease among all individual chemical components. Therefore, to alleviate the $PM_{2.5}$ pollution, measures focusing on sectors such as transportation, which is among the main sources of $NO_x$ emission, could be effective. Moreover, we find that a moderate 15% reduction of primary $PM_{2.5}$ emissions has resulted in a substantial reduction in ambient $PM_{2.5}$ concentrations, with the total concentration decreases in all primary $PM_{2.5}$ components exceeding that of nitrate. Therefore, a strengthened control on primary $PM_{2.5}$ emissions could be an effective strategy to sustainably mitigate $PM_{2.5}$ pollution. For $O_3$, reduction of $NO_x$ can effectively reduce the high $O_3$ concentrations in suburban areas, but may cause an increase of urban concentrations. A 70% VOC emission reduction is found to fully offset the urban $O_3$ increase caused by the lockdown. Therefore, the reduction in $NO_x$ emissions needs to be accompanied by a well-balanced reduction in VOC emissions to avoid the side effect on urban $O_3$ pollution.

### Data Availability Statement

The data from the California Air Resources Board (CARB) monitoring stations used in the present study can be obtained from https://www.arb.ca.gov/aqmis2/aqdselect.php. The meteorology observational data obtained from the National Climatic Data Center (NCDC) can be freely downloaded from ftp://ftp.ncdc.noaa.gov/pub/data/noaa/. Other data needed to support the findings of this study are in the manuscript and the Supplementary Information.

### Author contributions

BZ, YG, Y.Zhu, HS, and ZJ designed the research; HS, ZJ, and BZ conducted the research; ZJ, HS, BZ, and YG analyzed the results; KM and KWB provided the satellite-based emission data set; ZJ, BZ, HS, and YG wrote the paper with help from XL, Y.Zhang, KM, Y.Zhu, KWB, TS and KNL.

### Competing interests

The authors declare that they have no conflict of interest.

### Acknowledgments

B.Z. was partially supported by the DOE Atmospheric System Research (ASR) programme. Z.J. was partially supported by the National Natural Science Foundation of China (grant no. 41975178), the Youth Innovation Promotion Association of the Chinese Academy of Sciences, and a grant from State Key Laboratory of Resources and Environmental Information System. Y.G. and K.N.L. acknowledge the support by the NSF AGS-1660587, NASA TASNNP program, and NOAA-DOE CPT program. Y.G. and Y.Z. acknowledge the support by the LADWP Award# 20200288. Part of this work was conducted at the Jet Propulsion Laboratory, California Institute of Technology, under contract with NASA. K.M. and K.B. acknowledge the support by the NASA Atmospheric Composition: Aura Science Team Program (19-AURAST19-0044). We would like to acknowledge high-performance computing support from Cheyenne (https://doi.org/10.5065/D6RX99HX) provided by NCAR's Computational and Information Systems Laboratory, sponsored by the National Science Foundation. We also acknowledge the use of data products from the NASA Aura and EOS Terra and Aqua satellite missions from https://earthdata.nasa.gov. We also acknowledge the free use of the tropospheric $NO_2$ column data from the SCIAMACHY, GOME-2, and OMI sensors from http://www.qa4ecv.eu and from TROPOMI.

### Financial support

This study has been supported by the National Natural Science Foundation of China (grant no. 41975178), the Youth

Innovation Promotion Association of the Chinese Academy of Sciences, a grant from State Key Laboratory of Resources and Environmental Information System, the DOE Atmospheric System Research (ASR) programme, the NSF AGS-1660587, NASA TASNNP program, NOAA-DOE CPT program, the LADWP Award# 20200288, and the NASA Atmospheric

390 Composition: Aura Science Team Program (19-AURAST19-0044).

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

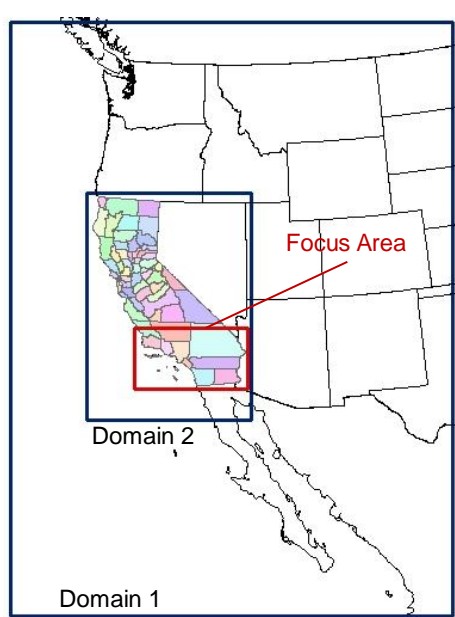


**Figure 1: Simulation domains of this study. The red rectangle denotes the area of southern California where most analyses in this study focus on.**

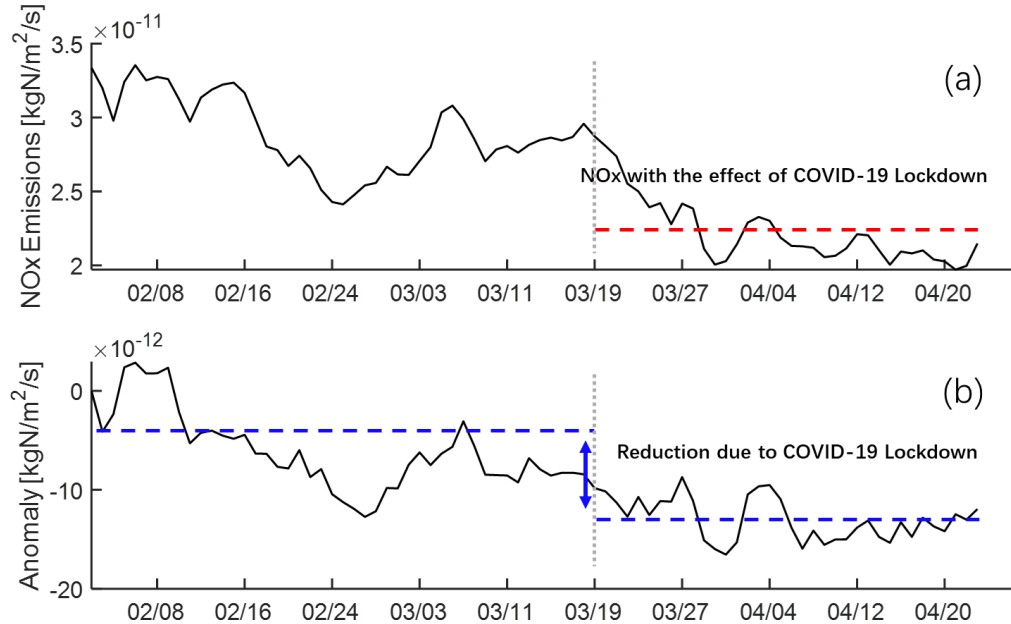

**Figure 2: Satellite-derived NO$_x$ emission estimates in southern California. (a) Daily NO$_x$ emissions from February 1 to April 23, 2020. The red line represents the average emissions during the period after March 19. (b) NO$_x$ emission changes due to the COVID-19 (i.e., the anomaly), which is quantified using the difference between the real-world NO$_x$ emissions and the emissions in a hypothetical scenario without considering the COVID-19. The emissions in the hypothetical scenario is estimated based on emission trends in prior years (2017–2019), using February 1 as a reference. The difference between two blue dashed lines represents the average reductions of NO$_x$ emissions induced by the COVID-19 lockdown measures that took effect on March 19. The local valley between February 24 and March 3 is caused by retrieval uncertainties caused by unfavorable meteorology conditions and is thus excluded when we estimate the average NO$_x$ emissions before the lockdown.**

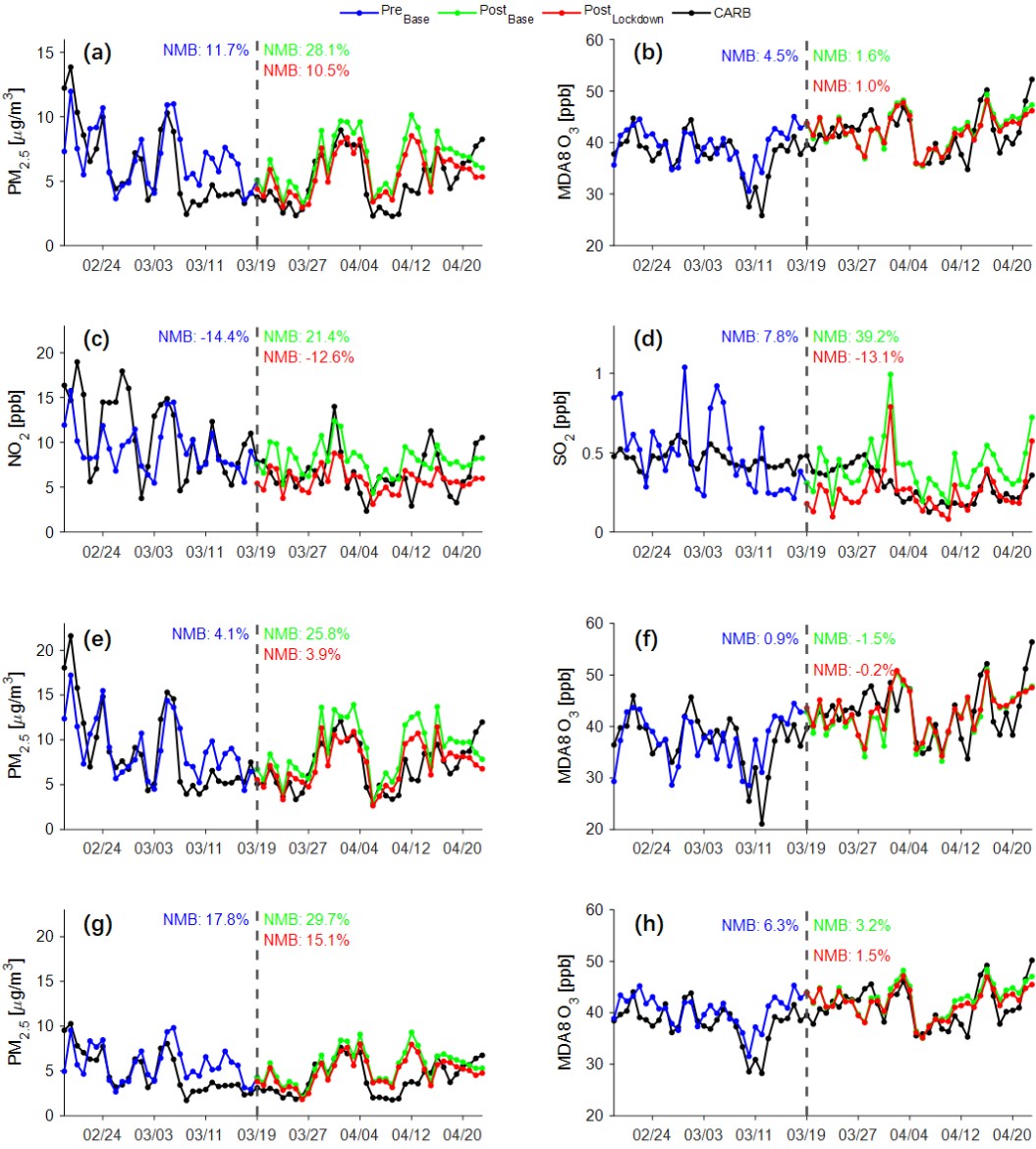

**Figure 3: Time series of observed and simulated concentrations of major pollutants. (a-d)** Time series of (a) PM$_{2.5}$, (b) MDA8 O$_3$, (c) NO$_2$, and (d) SO$_2$ averaged across all observational stations from CARB over southern California. **(e-f)** Time series of (e) PM$_{2.5}$ and (f) MDA8 O$_3$ across all stations over the urban areas of southern California. **(g-h)** The same as (e-f) but for the rural areas. Black lines are surface observations from the CARB network. Blue, green, and red lines are simulated air pollutant concentrations during the pre-lockdown period (February 18 to March 18) under the Base scenario (Pre$_{Base}$), during the lockdown period (March 19 to April 23) under the Base scenario (Post$_{Base}$), and during the lockdown period under the Lockdown scenario (Post$_{Lockdown}$). The definitions of the Base and Lockdown scenarios are summarized in Table 1. Normalized mean bias (NMB) is given by $= \sum_{i=1}^{N}(Var_m - Var_o)/\sum_{i=1}^{N} Var_o$, where N is the number of sites, Var$_m$ and Var$_o$ are modeled and observed concentrations, respectively.

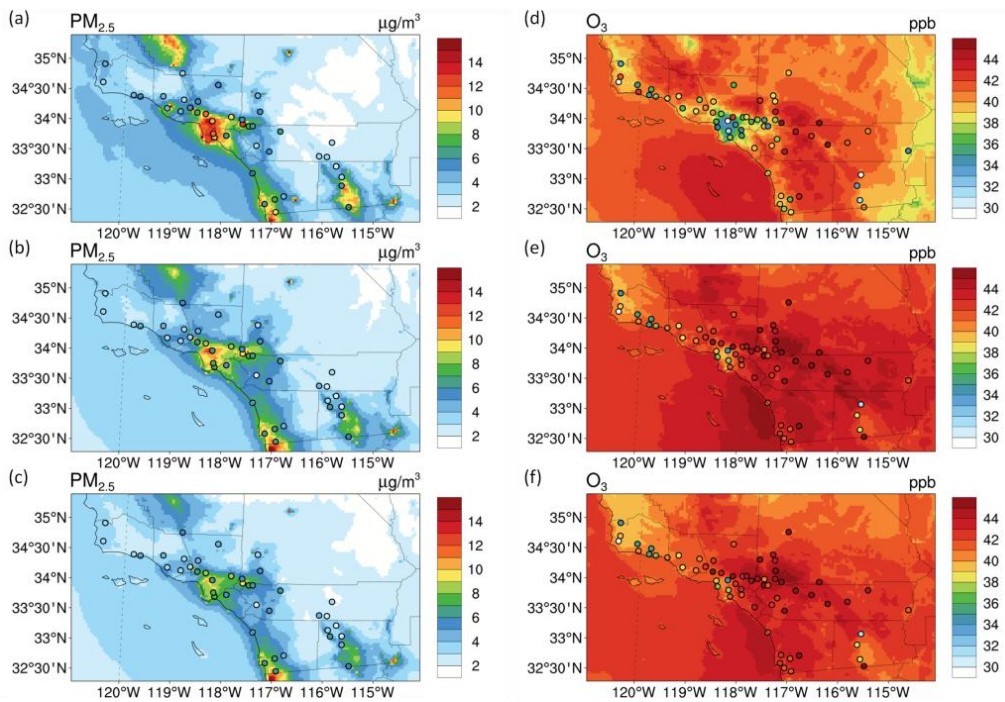

Figure 4: Overlay plots of the simulated (contour) and observed (circles) PM$_{2.5}$ and MDA8 O$_3$ concentrations in southern California. (a-c) are for PM$_{2.5}$ and (d-f) are for MDA8 O$_3$. (a, d) are for the pre-lockdown period (February 18 to March 18) under the Base scenario (Pre$_{Base}$); (b, e) are for the lockdown period (March 19 to April 23) under the Base scenario (Post$_{Base}$); (c, f) are for the lockdown period under the Lockdown scenario (Post$_{Lockdown}$).

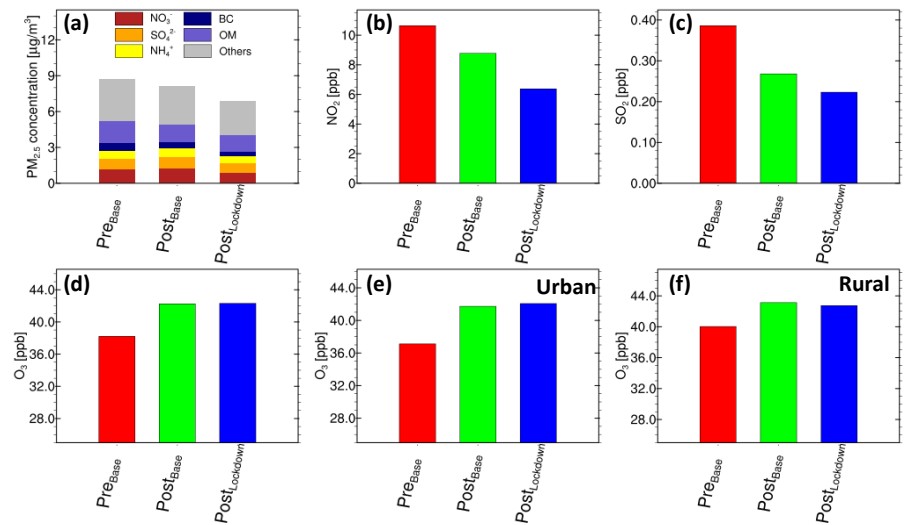

**Figure 5: Effects of emission reductions and meteorology conditions on air pollutants. (a-d) Population-weighted concentrations of simulated air pollutant concentrations in southern California: (a) $PM_{2.5}$ components; (b) $NO_2$; (c) $SO_2$; (d) MDA8 $O_3$ over southern California; (e) MDA8 $O_3$ over the urban areas of southern California; (f) MDA8 $O_3$ over the rural areas of southern California. Pre$_{Base}$, Post$_{Base}$, and Post$_{Lockdown}$ have the same meanings as in Fig. 3.**

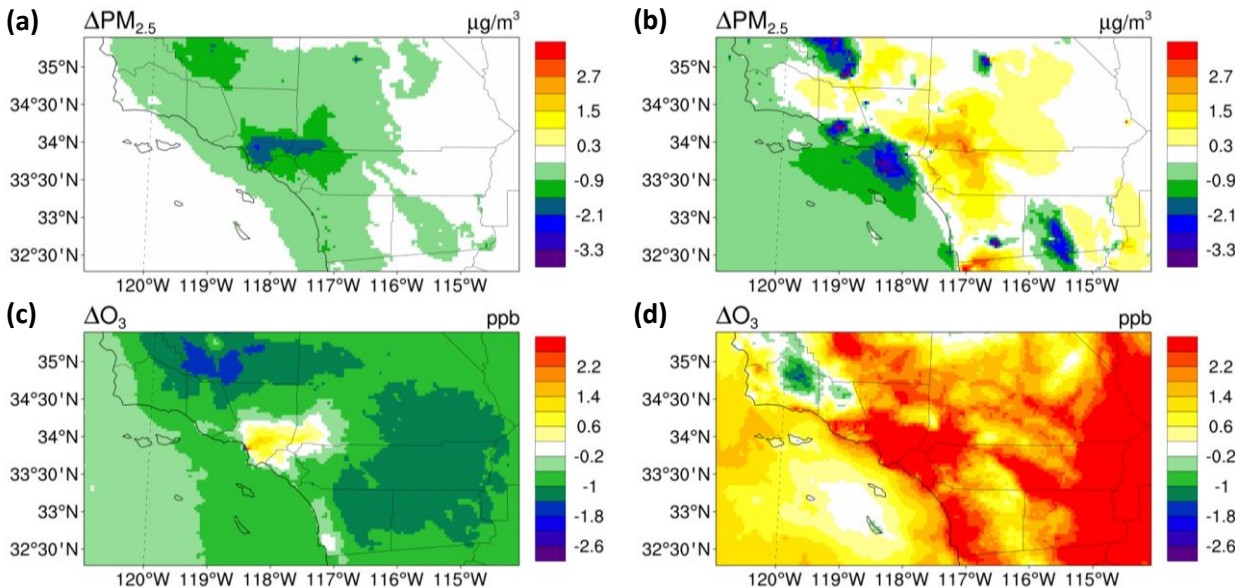


**Figure 6: Spatial distributions of the emission reductions and meteorology conditions effects on air pollutants. (a, c) emission reductions and (b, d) meteorology variations on (a,b) $PM_{2.5}$ and (c, d) MDA8 $O_3$ concentrations.**

=

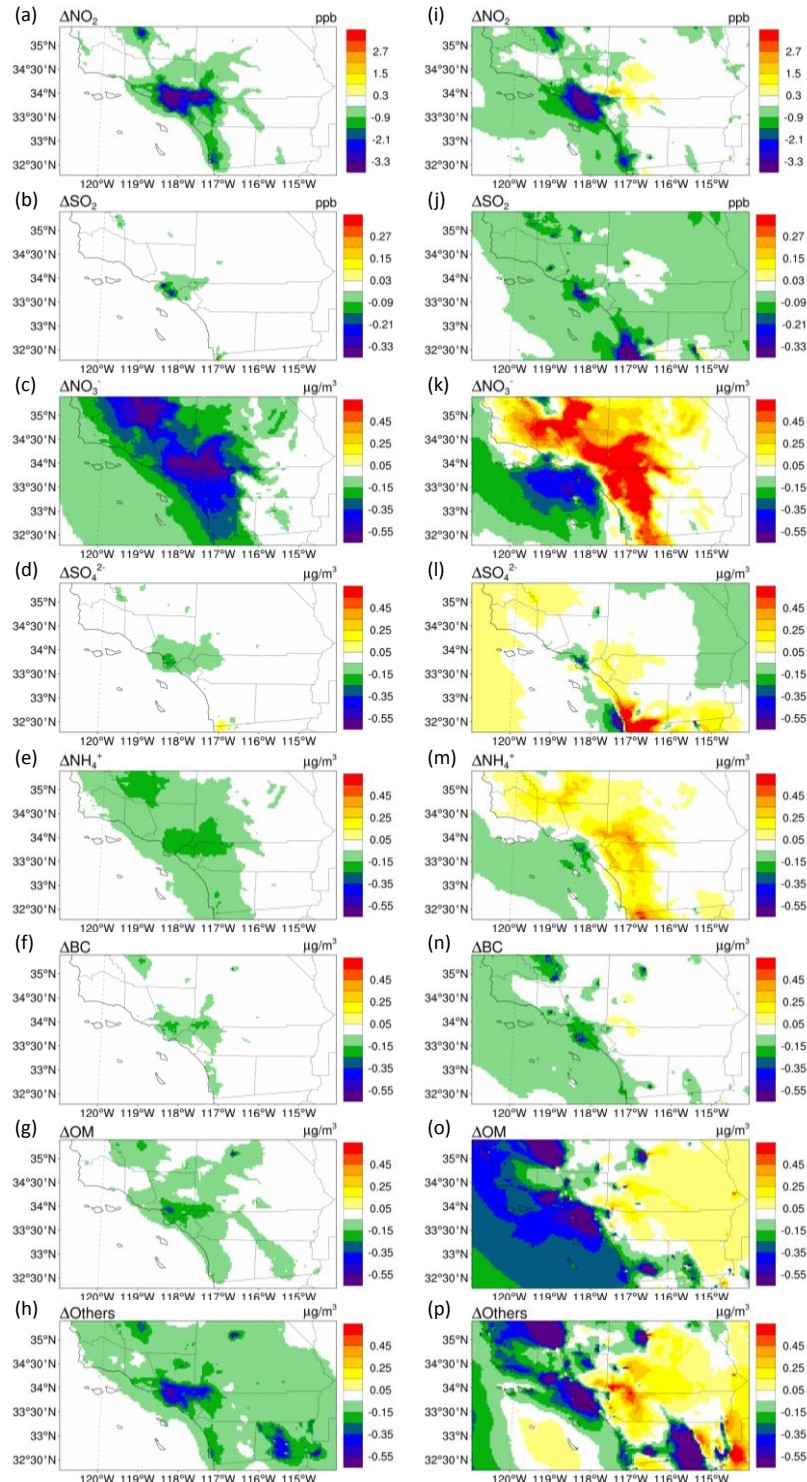

**Figure 7: The same as Figs. 6 but for NO₂, SO₂, and different PM₂.₅ chemical components.**

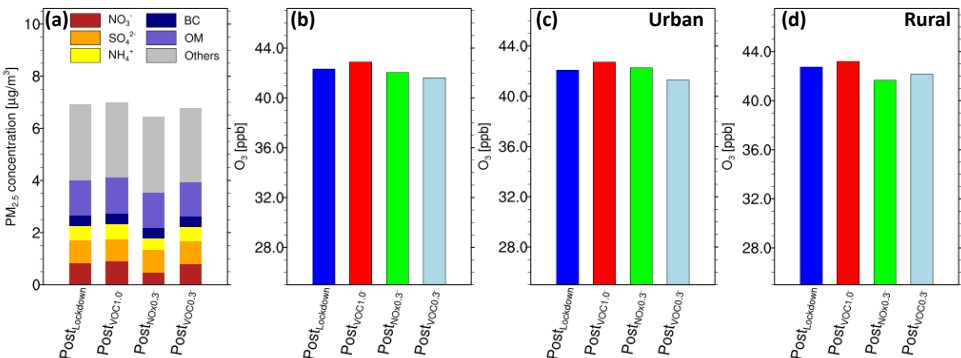

**Figure 8: Simulated population-weighted PM2.5 and O3 concentrations under three sensitivity scenarios (VOC1.0, NOx0.3 and VOC0.3) during the lockdown period (March 19 to April 23) over southern California. (a) PM2.5 components, (b) MDA8 O3, (c) MDA8 O3 over the urban areas, and (d) MDA8 O3 over the rural areas. The definitions of all scenarios are summarized in Table 1.**

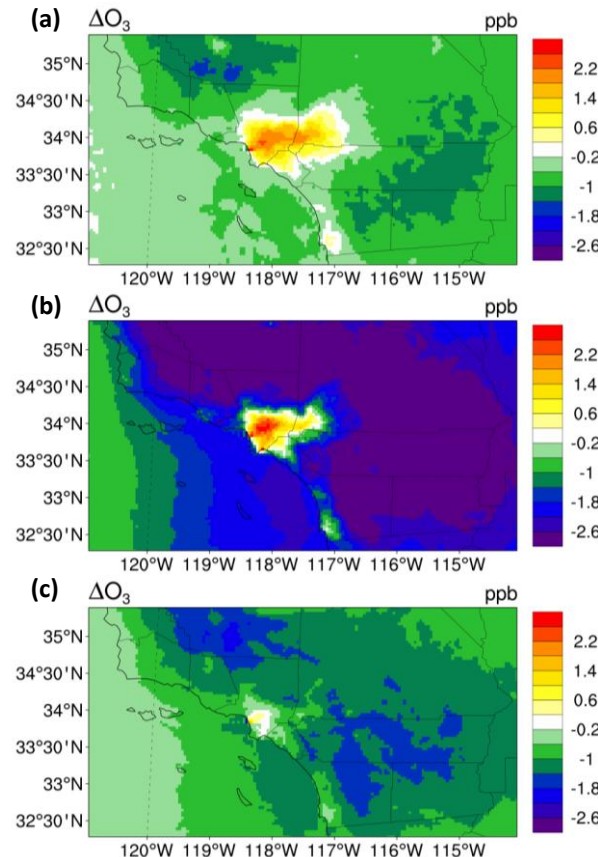


**Figure 9: Spatial distribution of the differences in MDA8 $O_3$ between the three sensitivity scenarios and the Base scenario: (a) VOC1.0 minus Base; (b) $NO_x0.3$ minus Base; (c) VOC0.3 minus Base. The definitions of all scenarios are summarized in Table 1.**

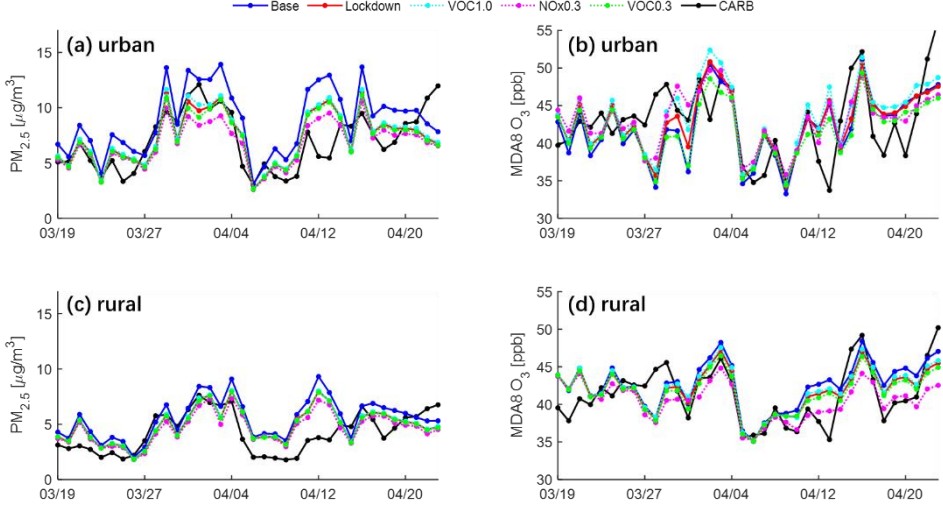

**Figure 10: Time series of simulated and observed PM$_{2.5}$ concentrations (a, c) and MDA8 O$_3$ concentrations (b, d) under several**
**sensitivity scenarios averaged across the CARB observational stations over the urban (a, b) and rural (c, d) areas of southern**
**California during the lockdown period (March 19 to April 23). Black lines are surface observations from CARB networks. Blue,**
**red, cyan, magenta, and green lines are simulated results for the Base, Lockdown, VOC1.0, NO$_x$0.3, and VOC0.3 scenarios. The**
**definitions of all scenarios are summarized in Table 1.**

**Table 1. Summary of model scenarios developed in this study.**

| Scenario | Definition |
|---|---|
| Base | This scenario uses the default CARB emission inventory without considering the emission reductions induced by the COVID-19 lockdown. It provides a baseline for evaluating the effect of COVID-19 lockdown on air quality. |
| Lockdown | This scenario adjusts the CARB emission inventory to account for the emission reductions due to the COVID-19 lockdown. The difference between "Base" and "Lockdown" represents the effect of the COVID-19 lockdown. |
| VOC1.0 | This scenario is the same as "Lockdown" except that the VOC emissions are kept at the level of the "Base" scenario. It is used to evaluate the relative contribution of VOC and NO$_x$ reductions to COVID-19 induced O$_3$ concentration changes. |
| NO$_x$0.3 | This scenario is the same as "Lockdown" except that the NO$_x$ emissions are further reduced to 30% of those in the "Base" scenario. It is used to assess the potential effects of strengthened NO$_x$ control measures. |
| VOC0.3 | This scenario is the same as "Lockdown" except that the VOC emissions are further reduced to 30% of those in the "Base" scenario. It is used to assess the potential effects of strengthened VOC control measures. |

**Table 2. The percentage of changes in air pollutant emissions during the COVID-19 lockdown relative to a hypothetical scenario without the lockdown in southern California.**

|  | VOC | CO | $NO_X$ | $SO_X$ | $PM_{10}$ | $PM_{2.5}$ | $NH_3$ |
|---|---|---|---|---|---|---|---|
| Onroad transportation | -50% | -51% | -39% | -35% | -44% | -42% | -51% |
| Off-road transportation | -30% | -30% | -30% | -30% | -30% | -30% | -30% |
| Aircraft | -70% | -70% | -70% | -70% | -70% | -70% | -70% |
| Power plants | -7% | -7% | -7% | -7% | -7% | -7% | -7% |
| Industrial | -15% | -15% | -15% | -15% | -15% | -15% | -15% |
| Residential | 10% | 10% | 10% | 10% | 10% | 10% | 10% |
| Commercial | -15% | -15% | -15% | -15% | -15% | -15% | -15% |
| Agriculture | 0% | 0% | 0% | 0% | 0% | 0% | 0% |
| Total | -21.1% | -35.7% | -28.3% | -18.5% | -9.7% | -15.0% | -16.1% |


**Table 3. Evaluation of meteorological simulation results as compared to observational data from the National Climatic Data Center.**

| Variable | Index | Value | Ref[a] | Variable | Index | Value | Ref[a] |
|---|---|---|---|---|---|---|---|
| Wind speed (m/s) | Mean observation | 3.92 |  | Temperature (K) | Mean observation | 287.48 |  |
|  | Mean simulation | 3.69 |  |  | Mean simulation | 287.21 |  |
|  | Mean Bias | -0.22 | ≤ ±0.5 |  | Mean Bias | -0.28 | ≤ ±0.5 |
|  | Gross error | 1.43 | ≤2 |  | Gross error | 1.76 | ≤2 |
|  | IOA[b] | 0.76 | ≥0.6 |  | IOA | 0.93 | ≥0.8 |
| Wind direction (deg) | Mean observation | 243.45 |  | Humidity (g/kg) | Mean observation | 6.41 |  |
|  | Mean simulation | 232.90 |  |  | Mean simulation | 6.16 |  |
|  | Mean Bias | 1.48 | ≤ ±10 |  | Mean Bias | -0.25 | ≤ ±1 |
|  | Gross error | 44.53 | ≤30 |  | Gross error | 0.83 | ≤2 |
|  |  |  |  |  | IOA | 0.84 | ≥0.6 |

[a] The reference values are taken from Emery et al. (2001).

[b] IOA = Index of Agreement.
