# Peer review of "Modeling the Impact of COVID-19 on Air Quality in Southern California: Implications for Future Control Policies"

_Atmospheric Chemistry and Physics, 2020_

## Referee Comment (RC1) · Anonymous Referee #3 · 28 Jan 2021

This manuscript investigated the air pollution during the COVID-19 lockdown period in Southern California. Using WRF-Chem modeling simulations, the authors found that PM2.5 concentrations decrease while O3 concentrations decrease/increase in rural/urban areas. This study suggests for Southern California control on primary PM2.5 emissions and balanced control on both NOx and VOCs emissions are needed to improve the air quality.

The text is concisely written and well documented. The topic is applicable for the Atmospheric Chemistry & Physics journal. However, the current manuscript misses detailed explanations and necessary analysis (please see the remarks below). First,

it is not clear how the emissions under COVID lockdown are projected. The authors listed several data sources and then 'scale' the 2012 CARB emissions, but did not show the details. The current manuscript also only listed the relative changes of each species by sector, but not the total change. Second, the modeling study completely ignore the non-linear O3-NOx-VOCs chemistry. The VOC-sensitive or NOx-sensitive regimes in southern CA could change under large emissions perturbations (i.e. 70% off in this case). More rigorous analysis is needed to support the conclusions. Third, the authors leave a lot of important information in the supplementary material. In my opinion, some of them should be moved to the main article.

In summary, the current manuscript shows important results but need further work. Major revisions as indicated in the comments and remarks below are needed before consideration of publication in ACP.

Detailed Remarks/Suggestions for Revision

Line 25: 'decrease' should be 'decreases'

Line 116: Table S2 is important, and needs to be moved into the main article. The author should also show the change of the total emissions because these sectors have different contributions. A figure could be added to the revised manuscript.

Line 117-120: How the authors estimate the emission changes from the fuel consumption? EPA uses the MOVES model to calculate the mobile emissions based on vehicle travel mileage, vehicle types, road types, and other factors. Which method is used here? Also in Table S2, the authors estimate the different reduction rates for onroad and off-road transportation. How it is computed?

Line 122-123: The assumption that the changes in power plant emissions are proportional to electricity demand in CA may ignore the impacts from interstate electricity transmission and the different responses from coal burning power plants, renewable energy sources such as wind and solar which might not change their outputs. EPA has

the CEMS program which is monitoring the power plant emissions of CO2, NOx, and SO2, which are more reliable for the modeling study.

Line 134-137: I am concerned about the approach using the top-down NOx emissions here. Usually there are substantial differences between the top-down emission products and bottom-up emission inventories, so it is hard to replace only NOx in the bottom-up emissions with a top-down estimate. Second, I don't understand how the COVID NOx emissions are calculated. Figures S2 says 'NOx emission changes due to the COVID-19, which is quantified using the difference between the real-world NOx emissions and the emissions in a hypothetical scenario without considering the COVID-19'. So the real-world NOx emissions are from the top-down products while the scenario without considering the COVID-19 is from the CARB emissions in 2012-2018 extrapolated to 2020? If that is the case, the authors should be prove that the bottom-up CARB emissions and top-down emission estimates are consistent spatially and quantitatively. More explanation is needed here. Lastly, adjust the NOx emissions are very important to this modeling study, so Figure S2 should be moved to the main article.

Line 137-138: I am more confused. FigS2 did show changes before and after 03/19. But as Goldberg (2020 mentioned in the introduction, this change may be caused by the natural variability of NOx (NO2 observed by satellite) due to change of temperature. Second, I am curious how the anomaly is calculated. To remove the seasonality, usually multi-year climatology is calculated first then the anomaly can be estimated. After reading the manuscript, I don't think the authors use this method. Detailed explanation is needed here.

Line 173-174: What is the criteria to define rural and urban here?

Line 180: Figure S3 shows the spatial performance of WRF-Chem, which should be moved to the main article. The figure is too smart to read. Can the authors add a scatter plot to show the model performance? It looks like WRF-Chem overestimate the

[Figure]

PM2.5 and O3 in LA basin during the post-lockdown periods, so it is not surprising the emission reductions can improve the model performance.

Line 190: Why use the population-weighted concentrations here? As mentioned above, the population-weighted concentrations will have more weights on populous LA basin area, where the baseline model did not have good performance.

Line 198: Are the soil NO emissions taken into account in WRF-Chem? With different meteorology, the natural NO emissions can play a role here.

Line 201-201: As mentioned above, need to update Table S2 to show the contribution to the total emissions.

Line 205: Again, the population-weighted concentration changes are mainly determined by the populous areas such as LA basin. Can the authors also show changes in mean concentrations?

Line 208-214: Fig2 e-f, how to separate the meteorological impacts and emissions impacts? The differences between 'Base' and 'Lockdown' contains impacts from both factors.

Line 226: What are the reductions in the primary PM2.5 emissions for other PM2.5?

Line248: Again, Table S1 should be moved to the main article so the readers can figure out the differences among these sensitivity experiments. Also, why the authors select coefficient 0.3 for the last two experiments? Is it estimated from the future regulations in CA?

Line 251-252: This statement ignored the non-linear chemistry of ozone.

Line 266-267: Same concern here, it is dangerous to use the reduction from NOx0.3 and VOC0.3 runs to conclude that the VOCs reduction can offset the NOx reduction because the nonlinear ozone chemistry is ignored here. With change of NOx and VOCs, the ozone production efficiency will change as well. I doubt in these two runs,

the ozone chemistry could shift into different regimes. Further analysis such as ozone isopleth diagram is needed here.

---

## Referee Comment (RC2) · Anonymous Referee #1 · 27 Feb 2021

The manuscript addresses a topical and critical science question, i.e. how air pollution responded to the abrupt changes in the human activities during the COVID-19 pandemic. A series of WRF-Chem simulations and ground-based observations are employed to understand the emission-pollution relationship in Southern California. The authors' effort of using observations of meteorology and pollution to evaluate and calibrate the WRF-Chem model is commendable. The finding about the dominant role of nitrate chemistry and primary PM emission in the observed PM2.5 reduction reinforce the importance of those two critical pathways for regional haze pollution in LA and lay out scientific foundation for future mitigation policy development. I recommend its publication with ACP, while I also have comments below for the authors to address.

[Figure]

1) The SI tables and figures contain very useful information about how the model is set up and the simulation results are evaluated. Since the main text only consists of three figures, I strongly suggest the authors move all SI tables and figures to the main text.

2) For Fig. 3a,b, better to separate the urban and rural areas, as they are in different ozone formation regimes. A simple regional average would largely reduce the ozone sensitivity to NOx or VOC in your plot.

3) Is the Miyazaki 2020c the same with Miyazaki 2020b?

4) The present study assesses the meteorological influence on the pollution concentrations by contrasting the lockdown and pre-lockdown time periods. It is a relatively crude way to achieve that objective, as the underlying assumption is that the pre-lockdown meteorology represents the climatological conditions during that time of the year. A more robust method is to conduct ensemble simulations over multiple years or use multi-year averaged meteorological conditions to serve as a reference state in the model (e.g. Le et al., 2020). The uncertainty here needs to be acknowledged.

5) The results in Table S3 are based on hourly or daily data? Over what area? A recent study by Rooney et al. (2020, ACP, p14597–14616, Fig. 5) found WRF-Chem tends to overpredict the nighttime low temperature in California. I wonder if the simulations presented here encounter the same issue.

---

## Author Comment (AC1) · 10 Apr 2021

The comment was uploaded in the form of a supplement: https://acp.copernicus.org/preprints/acp-2020-1197/acp-2020-1197-AC1-supplement.zip

---

## Author Comment (AC2) · 10 Apr 2021

The comment was uploaded in the form of a supplement:
https://acp.copernicus.org/preprints/acp-2020-1197/acp-2020-1197-AC2-supplement.zip

---

## Author Response (AR1)

The manuscript addresses a topical and critical science question, i.e. how air pollution responded to the abrupt changes in the human activities during the COVID-19 pandemic. A series of WRF-Chem simulations and ground-based observations are employed to understand the emission-pollution relationship in Southern California. The authors' effort of using observations of meteorology and pollution to evaluate and calibrate the WRF-Chem model is commendable. The finding about the dominant role of nitrate chemistry and primary PM emission in the observed PM2.5 reduction reinforce the importance of those two critical pathways for regional haze pollution in LA and lay out scientific foundation for future mitigation policy development. I recommend its publication with ACP, while I also have comments below for the authors to address.

We appreciate the reviewer's valuable comments, which have helped us improve the manuscript. We have carefully revised the manuscript according to these comments. Point-by-point responses are provided below. The reviewers' comments are in black and our responses are in blue.

1) The SI tables and figures contain very useful information about how the model is set up and the simulation results are evaluated. Since the main text only consists of three figures, I strongly suggest the authors move all SI tables and figures to the main text.

R: Done. Thank you!

2) For Fig. 3a,b, better to separate the urban and rural areas, as they are in different

ozone formation regimes. A simple regional average would largely reduce the ozone sensitivity to NOx or VOC in your plot.

R: Following the reviewer's suggestion, we have added two panels showing the population-weighted concentrations of simulated MDA8 O₃ over the urban and rural areas in Fig. 5 and Fig. 8 (Fig. 2 and Fig. 3 in the original manuscript). We do not show the urban and rural PM₂.₅ concentrations separately because the patterns of PM₂.₅ concentration changes are similar in these two settings.

[Figure]

**Figure 5:** Effects of emission reductions and meteorology conditions on air pollutants. (a-d) Population-weighted concentrations of simulated air pollutant concentrations in southern California: (a) PM₂.₅ components; (b) NO₂; (c) SO₂; (d) MDA8 O₃ over southern California; (e) MDA8 O₃ over the urban areas of southern California; (f) MDA8 O₃ over the rural areas of southern California. Pre_Base, Post_Base, and Post_Lockdown have the same meanings as in Fig. 3.

[Figure]

**Figure 8:** Simulated population-weighted PM₂.₅ and O₃ concentrations under three sensitivity scenarios (VOC1.0, NOx0.3 and VOC0.3) during the lockdown period (March 19 to April 23) over southern California. (a) PM₂.₅ components, (b) MDA8 O₃, (c) MDA8 O₃ over the urban areas, and (d) MDA8 O₃ over the rural

areas. The definitions of all scenarios are summarized in Table 1.

3) Is the Miyazaki 2020c the same with Miyazaki 2020b?

R: Thank you! We have removed Miyazaki 2020c.

4) The present study assesses the meteorological influence on the pollution concentrations by contrasting the lockdown and pre-lockdown time periods. It is a relatively crude way to achieve that objective, as the underlying assumption is that the prelockdown meteorology represents the climatological conditions during that time of the year. A more robust method is to conduct ensemble simulations over multiple years or use multi-year averaged meteorological conditions to serve as a reference state in the model (e.g. Le et al., 2020). The uncertainty here needs to be acknowledged.

R: Thank you for the suggestion. In fact, we did not assume that the pre-lockdown meteorology represents the climatological conditions since it was our intention to examine the relative contribution of meteorological variations to the changes in population-weighted air pollutant concentrations before and after the lockdown, instead of the changes relative to the climatological conditions. However, we acknowledge that it is more meaningful and informative to assess the meteorological effect by conducting ensemble simulations over multiple years or use multi-year averaged meteorological conditions to serve as a reference state (Le et al., 2020), which warrants further studies in the future. We have mentioned this in the revised manuscript (Lines 225-229).

5) The results in Table S3 are based on hourly or daily data? Over what area? A recent

study by Rooney et al. (2020, ACP, p14597–14616, Fig. 5) found WRF-Chem tends to overpredict the nighttime low in California. I wonder if the simulations presented here encounter the same issue.

R: The results in Table S3 (now Table 3) are based on hourly data for 82 sites distributed in southern California (the red rectangle in now Fig. 1). As shown in the following figures which represent the temperature during the nighttime and daytime, respectively, we didn't find that WRF-Chem tends to overpredict the nighttime low temperature in our study.

[Figure]

Figure: Comparison of NCDC temperature observations versus WRF-Chem simulations

**Anonymous Referee #3**

This manuscript investigated the air pollution during the COVID-19 lockdown period in Southern California. Using WRF-Chem modeling simulations, the authors found that $PM_{2.5}$ concentrations decrease while $O_3$ concentrations decrease/increase in rural/ urban areas. This study suggests for Southern California control on primary $PM_{2.5}$ emissions and balanced control on both $NO_x$ and VOCs emissions are needed to improve the air quality. The text is concisely written and well documented. The topic is applicable for the Atmospheric Chemistry & Physics journal. However, the current manuscript misses

detailed explanations and necessary analysis (please see the remarks below). First, it is not clear how the emissions under COVID lockdown are projected. The authors listed several data sources and then 'scale' the 2012 CARB emissions, but did not show the details. The current manuscript also only listed the relative changes of each species by sector, but not the total change. Second, the modeling study completely ignore the non-linear $O3$-$NO_x$-VOCs chemistry. The VOC-sensitive or NOx-sensitive regimes in southern CA could change under large emissions perturbations (i.e. 70% off in this case). More rigorous analysis is needed to support the conclusions. Third, the authors leave a lot of important information in the supplementary material. In my opinion, some of them should be moved to the main article.

In summary, the current manuscript shows important results but need further work. Major revisions as indicated in the comments and remarks below are needed before consideration of publication in ACP.

We thank the reviewer for the valuable comments. We have carefully revised the manuscript according to these comments. In particular, we have added the details about how the emissions under COVID lockdown are projected (see our responses to the reviewer's comments on Line 117-120, 122-123, and 226 below, as well as Lines 122-141 and 266-270 in the revised manuscript and Lines 35-50 in the SI). We have also added the total emission changes in Table S2 (now Table 2). We have considered the nonlinearity in $O_3$ chemistry when we design and interpret the sensitivity scenarios (see our responses to the reviewer's comments on Line 251-252 and Line 266-267). Finally, we have moved most of the tables and figures in the Supplementary Material to the main

text. More detailed point-by-point responses are provided below. The reviewers' comments are in black and our responses are in blue.

Detailed comments:

Line 25: 'decrease' should be 'decreases'

R: Done. Thank you! (See Line 27).

Line 116: Table S2 is important, and needs to be moved into the main article. The author should also show the change of the total emissions because these sectors have different contributions. A figure could be added to the revised manuscript.

R: Thank you for your comments. We have added the total emission changes in Table S2 (now Table 2). We have also added the following figure (Fig. S2) to show the emission changes with and without the lockdown.

Table 2. The percentage of changes in air pollutant emissions during the COVID-19 lockdown relative to a hypothetical scenario without the lockdown in southern California.

| | VOC | CO | $NO_X$ | $SO_X$ | $PM_{10}$ | $PM_{2.5}$ | $NH_3$ |
|---|---|---|---|---|---|---|---|
| Onroad transportation | -50% | -51% | -39% | -35% | -44% | -42% | -51% |
| Off-road transportation | -30% | -30% | -30% | -30% | -30% | -30% | -30% |
| Aircraft | -70% | -70% | -70% | -70% | -70% | -70% | -70% |
| Power plants | -7% | -7% | -7% | -7% | -7% | -7% | -7% |
| Industrial | -15% | -15% | -15% | -15% | -15% | -15% | -15% |
| Residential | 10% | 10% | 10% | 10% | 10% | 10% | 10% |
| Commercial | -15% | -15% | -15% | -15% | -15% | -15% | -15% |
| Agriculture | 0% | 0% | 0% | 0% | 0% | 0% | 0% |
| Total | -21.1% | -35.7% | -28.3% | -18.5% | -9.7% | -15.0% | -16.1% |

[Figure]

Figure S2: Air pollutant emissions in southern California with (red) and without (blue) the COVID-19 lockdown.

Line 117-120: How the authors estimate the emission changes from the fuel consumption? EPA uses the MOVES model to calculate the mobile emissions based on vehicle travel mileage, vehicle types, road types, and other factors. Which method is used here? Also in Table S2, the authors estimate the different reduction rates for onroad and off-road transportation. How it is computed?

R: Thanks for your comments. In our study, we obtain anthropogenic emissions in California without the influence of COVID-19 lockdown measures from the California Air Resources Board (CARB). In the CARB emission inventory, emissions from the transportation sector were estimated using the EMission FACtor (EMFAC) model. The EMFAC and MOVES models use a similar concept to estimate emissions based on vehicle activities, base emission rates, and a series of adjustment factors (Vallamsundar et al., 2011). Differences between MOVES and EMFAC are mainly reflected in how vehicle activities are quantified, how emission rates are measured, and how vehicle

activities and emission rates are paired spatially and temporally.

For the lockdown period, we are not able to estimate the changes in transportation emissions using the EMFAC model because the detailed input data needed by the model are not available during this period. Thus, we use a simplified method to obtain the reduction rates for onroad and off-road sources due to the COVID-19 lockdown. Specifically, we assume the reduction rates in gasoline and diesel vehicle emissions in the onroad sector to be the same as the reduction rates in gasoline and diesel production from the pre-lockdown period to the lockdown period, as documented by California Energy Commission's "Weekly Fuels Watch Reports" (California Energy Commission, 2020b). We then estimate the reduction rates in total emissions from the onroad sector based on the relative fractions of gasoline and diesel vehicle emissions reported by the CARB emission inventory. Since the off-road sector involves few gasoline vehicles, we assume the reduction rates in off-road emissions to be the same as the reduction rate in diesel production.

We have added the above description of how the emission changes in Lines 124-133.

Vallamsundar, S. and Lin, J., MOVES versus MOBILE: comparison of greenhouse gas and criterion pollutant emissions. Transportation research record, 2233(1), 27-35, 2011.

Line 122-123: The assumption that the changes in power plant emissions are proportional to electricity demand in CA may ignore the impacts from interstate electricity transmission and the different responses from coal burning power plants, renewable energy sources such as wind and solar which might not change their outputs. EPA has the CEMS

program which is monitoring the power plant emissions of CO2, NOx, and SO2, which are more reliable for the modeling study.

R: We thanks the reviewer for the constructive suggestions. We checked the change of $CO_2$, $NO_x$, and $SO_2$ emissions from power plants measured by the CEMS. The time series of $SO_2$ and $CO_2$ emissions in southern California during the pre-lockdown and lockdown periods are shown in the following figures ($NO_x$ emissions are not available during this period). We can see that the CEMS-based $SO_2$ emissions have a strong day-to-day variation, making it difficult to achieve an accurate estimate of the COVID-19 related emission changes. The average $SO_2$ emission decreases by 39% between the pre-lockdown and lockdown periods defined in this study, larger than the reduction rate estimated based on electricity demand (7%). However, it is noted that the above CEMS-based reduction rate is also subject to large uncertainty due to the strong fluctuation of emission rates.

We then examined the potential impact of this difference on our results. As reported in the CARB emission inventory (CARB, 2021), the emissions of VOC, CO, $NO_x$, and $PM_{10}$ from power plants account for less than 1% of the total emissions, and the emissions of $SO_2$, $NH_3$, and $PM_{2.5}$ all account for less than 3%. For this reason, the different emission reduction rates estimated based on the CEMS and electricity demand will translate into less than 1% difference in the total emissions of any pollutant (ranged from 0.05% to 1%) , which is expected to have a limited effect on the simulation results of mean air pollutant concentrations in southern California.

We have added the above description in Lines 137-141 in the revised manuscript and

[Figure]

Figure S3: The power plant emissions of $SO_2$ and $CO_2$ in southern California measured by the CEMS before and during the COVID-19 lockdown.

Line 134-137: I am concerned about the approach using the top-down NOx emissions here. Usually there are substantial differences between the top-down emission products and bottom-up emission inventories, so it is hard to replace only $NO_x$ in the bottom-up emissions with a top-down estimate. Second, I don't understand how the COVID $NO_x$ emissions are calculated. Figures S2 says 'NO$_x$ emission changes due to the COVID-19, which is quantified using the difference between the real-world $NO_x$ emissions and the emissions in a hypothetical scenario without considering the COVID-19'. So the real-world

NO$_x$ emissions are from the top-down products while the scenario without considering the COVID-19 is from the CARB emissions in 2012- 2018 extrapolated to 2020? If that is the case, the authors should prove that the bottom-up CARB emissions and top-down emission estimates are consistent spatially and quantitatively. More explanation is needed here. Lastly, adjust the NO$_x$ emissions are very important to this modeling study, so Figure S2 should be moved to the main article.

R: We did not replace the NO$_x$ emissions in the bottom-up inventory with the top-down estimates. Instead, we estimated the COVID-19 related emission reductions based on the top-down and bottom-up NO$_x$ inventories separately, and subsequently compared the two to prove the reliability of our bottom-up emissions.

In the lines the reviewer refers to, both the "real-world NO$_x$ emissions" and the "emissions in a hypothetical scenario" are derived from the top-down NO$_x$ emission inventory based on satellite measurements. Here we try to explain the method more clearly with the following conceptual figure. In the figure, t1 is a reference time (February 1 in this work) and t2 is a time during the lockdown period. The COVID-19 induced emission changes during the lockdown period (t2) are calculated from the difference between the real-world emissions (red solid line) and the hypothetical emissions without considering the COVID-19 (red dashed line). We introduce the hypothetical scenario because the emissions at t1 and t2 would be different due to natural variability, even if the COVID-19 lockdown did not take place. The hypothetical emission change between t1 and t2 in 2020 is estimated using the average emission changes in the corresponding periods during the prior three years (2017–2019).

We have moved Figure S2 to the main text (now Figure 2) and added the above descriptions in Lines 152-156.

[Figure]

Figure: The concept of the estimation of $NO_x$ emission ($ENO_x$) reductions due to the COVID-19 based on the top-down emission inventory.

Line 137-138: I am more confused. FigS2 did show changes before and after 03/19. But as Goldberg (2020 mentioned in the introduction, this change may be caused by the natural variability of $NO_x$ ($NO_2$ observed by satellite) due to change of temperature. Second, I am curious how the anomaly is calculated. To remove the seasonality, usually multi-year climatology is calculated first then the anomaly can be estimated. After reading the manuscript, I don't think the authors use this method. Detailed explanation is needed here.

R: The $NO_x$ data analyzed in Goldberg et al. (2020) are the column-integrated density of $NO_2$, which depends strongly on meteorological conditions. In contrast, Fig. S2 (now Fig. 2) shows $NO_x$ emissions derived from a state-of-the-art technique that combines satellite data and a global chemical transport model (Miyazaki et al., 2020a). Compared with $NO_2$

column density, $NO_x$ emissions are much less affected by meteorology.

As explained in the last comments, the emission changes due to the COVID-19 (i.e., the anomaly) are calculated from the difference between the real-world emissions and the hypothetical emissions without considering the COVID-19 based on the top-down $NO_x$ inventory. The hypothetical emissions do represent the climatological conditions since they are calculated based on the average emission changes in the prior three years (2017-2019).

Line 173-174: What is the criteria to define rural and urban here?

R: According to Ratcliffe et al. (2016), to be classified as "urban", an area in the U.S. needs to have a population density of 1,000 people per square mile, i.e., about 6000 people per 4 km×4 km model grid. As we focus our analysis on southern California, one of the most densely populated areas in the U.S., we use a higher population density threshold of 30,000 people per model grid to better distinguish areas with different photochemistry regimes. We have described the criteria and included the following figure in the revised manuscript (Lines 96-101 and Fig. S1).

[Figure]

Figure S1: The population density in the area this study focuses on.

Line 180: Figure S3 shows the spatial performance of WRF-Chem, which should be moved to the main article. The figure is too smart to read. Can the authors add a scatter plot to show the model performance? It looks like WRF-Chem overestimate the PM$_{2.5}$ and O$_3$ in LA basin during the post-lockdown periods, so it is not surprising the emission reductions can improve the model performance.

R: Following the reviewer's suggestion, we have moved Figure S3 to the main text (now Fig. 4) and added the following scatter plot in SI. As the reviewer pointed out, without considering the emission reduction, the simulation overestimates the PM$_{2.5}$ and O$_3$ concentrations during the post-lockdown period in the Los Angeles basin (the relatively high PM$_{2.5}$ concentrations and low O$_3$ concentrations in Fig. 4b and e). The PM$_{2.5}$ simulation results are generally improved after considering the emission reduction, though certain biases still exist (Fig. S4c).

[Figure]

Figure S4: Scattergrams of the simulated and observed monthly average PM$_{2.5}$ and MDA8 O$_3$ concentrations in southern California. (a-c) are for PM$_{2.5}$ and (d-f) are for MDA8 O$_3$. (a,

d) are for the pre-lockdown period (February 18 to March 18) under the Base scenario

(Pre$_{Base}$); (b, e) are for the lockdown period (March 19 to April 23) under the Base scenario

(Post$_{Base}$); (c, f) are for the lockdown period under the Lockdown scenario (Post$_{Lockdown}$).

Line 190: Why use the population-weighted concentrations here? As mentioned above,

the population-weighted concentrations will have more weights on populous LA basin

area, where the baseline model did not have good performance.

R: We used population-weighted concentrations because they are more relevant to the

health impacts of air pollutants (PM$_{2.5}$ and O$_3$), the mitigation of which is an ultimate goal

of controlling air pollution. We have also added Fig. S5 to show the mean concentrations

in southern California in SI. We have mentioned this in the revised manuscript (Lines

212-215).

Line 198: Are the soil NO emissions taken into account in WRF-Chem? With different

meteorology, the natural NO emissions can play a role here.

R: The soil NO$_x$ emissions haven't been taken into account in our WRF-Chem simulation.

According to Guo et al. (2020), the total soil NO$_x$ emissions in California account for about

1.1% of the state's total anthropogenic NO$_x$ emissions (CARB, 2017). Soil NO$_x$ fluxes are

highly variable across the state, depending on land-use patterns. California can be divided

into three broad soil NO$_x$ emission zones: the high emission zone in the Central Valley

covered by cropland, the low emission zone in the southeast region dominated by

shrubland, and the intermediate emission zone covered by grassland and forest for the

rest of the state. Most areas over southern California belong to the low and intermediate emission zones, so soil $NO_x$ emissions are generally low in the regions our study focuses on.

In addition, our study focuses on the impact of emission reductions on air quality during the COVID-19 lockdown period, which is quantified by comparing the concentrations in two scenarios between which the only difference is anthropogenic emissions (i.e., $Post_{Lockdown}$ and $Post_{Base}$). The meteorological conditions in the two scenarios are almost the same, leading to roughly the same soil emissions. Therefore, even if the soil $NO_x$ emissions affect air quality in the Base scenario to some extent, they are expected to play a very small role in the impact of emission reductions on air quality during the lockdown period and wouldn't change the results and findings of this study. We have added the preceding descriptions in the revised manuscript (Lines 236-242).

Line 201-201: As mentioned above, need to update Table S2 to show the contribution to the total emissions.

R: We have added the total emission changes in Table S2 (now Table 2). We have also added Fig. S2 in SI to show the emission changes with and without the lockdown.

Line 205: Again, the population-weighted concentration changes are mainly determined by the populous areas such as LA basin. Can the authors also show changes in mean concentrations?

R: Following the reviewer's suggestion, we have added the following figure which shows

changes in mean concentrations.

[Figure]

Figure S5: Mean concentrations of simulated air pollutant concentrations in southern California: (a) PM$_{2.5}$ components; (b) MDA8 O$_3$; (c) NO$_2$; (d) SO$_2$. Pre$_{Base}$, Post$_{Base}$, and Post$_{Lockdown}$ have the same meanings as in Fig. 3.

Line 208-214: Fig2 e-f, how to separate the meteorological impacts and emissions impacts? The differences between 'Base' and 'Lockdown' contains impacts from both factors.

R: The simulations of the Base scenario during the lockdown and pre-lockdown periods (Post$_{Base}$ and Pre$_{Base}$) both use the emission inventories without considering the COVID-19 induced emission reductions. The differences between Post$_{Base}$ and Pre$_{Base}$ can be regarded as the impact of meteorology variations.

The simulations of the Base and Lockdown scenarios during the lockdown period (Post$_{Base}$ and Post$_{Lockdown}$) have the same model configurations and inputs (same large scale meteorological conditions) except for different emission inventories. The concentration differences between the two scenarios during the lockdown period (Post$_{Lockdown}$ – Post$_{Base}$) represent the effect of anthropogenic emission reductions. Strictly

speaking, while the large-scale meteorological fields are the same in $Post_{Base}$ and

$Post_{Lockdown}$, the different emission inputs could cause small differences in regional

meteorology fields through the interactions between air pollutants and meteorology. Such

a meteorology perturbation is considered to be part of the emission reduction effect

because it is fundamentally caused by emission reductions.

We have added the above description in Lines 216-225.

Line 226: What are the reductions in the primary $PM_{2.5}$ emissions for other $PM_{2.5}$?

R: The sector-specific relative changes of "other $PM_{2.5}$" (primary $PM_{2.5}$ except for EC and

primary OM) emissions are assumed to be the same as the total primary $PM_{2.5}$, as

summarized in Table S2 (now Table 2). For the total emissions of all sectors, the

reduction in "other $PM_{2.5}$" emissions is 13.5%, slightly smaller than the reduction in total

primary $PM_{2.5}$ since different chemical components have different sectoral distributions.

We have mentioned this in the revised manuscript (Lines 266-270).

Line248: Again, Table S1 should be moved to the main article so the readers can figure

out the differences among these sensitivity experiments. Also, why the authors select

coefficient 0.3 for the last two experiments? Is it estimated from the future regulations in

CA?

R: We have moved Table S1 to the main manuscript (now Table 1).

We select an emission ratio of 0.3 (70% reduction) to represent the potential impact of

highly stringent control policies in the future. According to Amann et al. (2020), a 70%

reduction is close to the maximum reductions in $NO_x$ and VOC emissions that could be achieved through the full implementation of technologically and economically feasible control measures.

We have added the above description in Lines 303-306.

Line 251-252: This statement ignored the non-linear chemistry of ozone.

R: We have revised this sentence to "This scenario, in combination with the Base and Lockdown scenarios, can be used to evaluate the response of $O_3$ concentrations if the COVID-19 induced emission reductions of $NO_x$ and VOC were implemented in sequence." in Lines 295-297.

Line 266-267: Same concern here, it is dangerous to use the reduction from $NO_x0.3$ and VOC0.3 runs to conclude that the VOCs reduction can offset the $NO_x$ reduction because the nonlinear ozone chemistry is ignored here. With change of $NO_x$ and VOCs, the ozone production efficiency will change as well. I doubt in these two runs, the ozone chemistry could shift into different regimes. Further analysis such as ozone isopleth diagram is needed here.

R: Thank you for your comment. In fact, we have considered the nonlinearity in $O_3$ chemistry when we design and interpret these sensitivity scenarios. The VOC0.3 scenario is the same as the Lockdown scenario except that the anthropogenic VOC emissions are further reduced to 30% of those in the Base scenario. While a $NO_x$ emission reduction might cause an increase in $O_3$ concentration, a VOC reduction generally leads to a

monotonous reduction of $O_3$ concentrations regardless of the $O_3$ formation regime, as indicated by the classical $O_3$ EKMA isopleth (Figure 6-1 of National Research Council (1991) or Figure 3.2.1 of Donahue (2018)) as well as some recent studies in southern California (Fujita et al., 2013; Collet et al., 2018; Qian et al., 2019). We find that in the VOC0.3 scenario, there is almost no $O_3$ concentration increase relative to the Base scenario, in contrast to a significant urban $O_3$ increase in the Lockdown scenario (Fig. 9c). This means that a 70% reduction in anthropogenic VOC can offset the increases in $O_3$ caused by the 28.3% $NO_x$ reduction during the lockdown. Note that we are specifically looking at the extent of VOC emission reductions that are needed to offset the 28.3% $NO_x$ reduction caused by the lockdown, which minimizes the complexity due to the nonlinear $O_3$ responses when the $NO_x$ emissions are changing simultaneously. We have included the above explanations in the revised manuscript (Lines 314-323).